# EfficientPhys: Enabling Simple, Fast and Accurate Camera-Based Cardiac Measurement

## Abstract

Camera-based physiological measurement is a growing field with neural models providing state-the-art-performance. Prior research have explored various "end-to-end" models; however these methods still require several preprocessing steps. These additional operations are often non-trivial to implement making replication and deployment difficult and can even have a higher computational budget than the "core" network itself. In this paper, we propose two novel and efficient neural models for camera-based physiological measurement called EfficientPhys that remove the need for face detection, segmentation, normalization, color space transformation or any other preprocessing steps. Using an input of raw video frames, our models achieve state-of-the-art accuracy on three public datasets. We show that this is the case whether using a transformer or convolutional backbone. We further evaluate the latency of the proposed networks and show that our most light weight network also achieves a 33% improvement in efficiency.

## 1 Introduction

Camera-based physiological measurement is a non-contact approach for capturing cardiac signals via light reflected from the body. The most common such signal is the blood volume pulse (BVP) measured via the photoplethysmogram (PPG). From this, heart rate (Takano & Ohta, 2007; Verkruysse et al., 2008), respiration rate (Poh et al., 2010) and pulse transit times Shao et al. (2014) can be derived. Furthermore, there is promising evidence that the PPG signals be be used to measure signs of arterial disease Takazawa et al. (1998). Neural models are the current state-of-the-art in this domain (Chen & McDuff, 2018a; Liu et al., 2020a; 2021d). These networks can learn strong feature representations and effectively disentangle the subtle changes in pixels due to underlying physiological processes from those due to body motions, lighting changes and other sources of "noise".

While prior research has framed architectures as "end-to-end" methods, those that achieve state-of-the-art performance actually require several preprocessing steps before data is input into the network. For example, Chen & McDuff (2018a) and Liu et al. (2020a) use hand-crafted normalized difference frames and normalized appearance frames as input to their convolutional attention network. Niu et al. (2020) and Lu et al. (2021) use a complex schema to create feature maps called "MSTmap". This process includes facial landmark detection, extractions of several regions of interest (ROI) using these landmarks, and then averaging pixel values in both the RGB and YUV color spaces.

These preprocessing steps have several drawbacks: 1) They make assumptions about optimal normalization or representation without allowing the network to learn these features in a data-driven manner. 2) They are computationally costly and in many cases add a significant number of operations to the video processing pipeline. There are several reasons why we would prefer to run camera-based physiological sensing on-device: preserving privacy, analyzing raw video (i.e., not compressed) and saving data costs and bandwidth. Therefore, any additional computation needs to be justified by improving model accuracy, otherwise it is considerably disadvantageous. Moreover, since camera-based physiological sensing is a privacy sensitive application, it is preferred to store the data at local devices instead of streaming the video and physiological data to cloud. The overhead from processing is not affordable if we aim to make the system accessible to low-end mobile devices. 3) Many of these steps are non-trivial to implement and optimize in and of themselves. This makes it harder to deploy real-time systems and to replicate the implementation on different platforms. For instance, implementing existing methods on Android, iOS, or in JavaScript requires a significant amount of

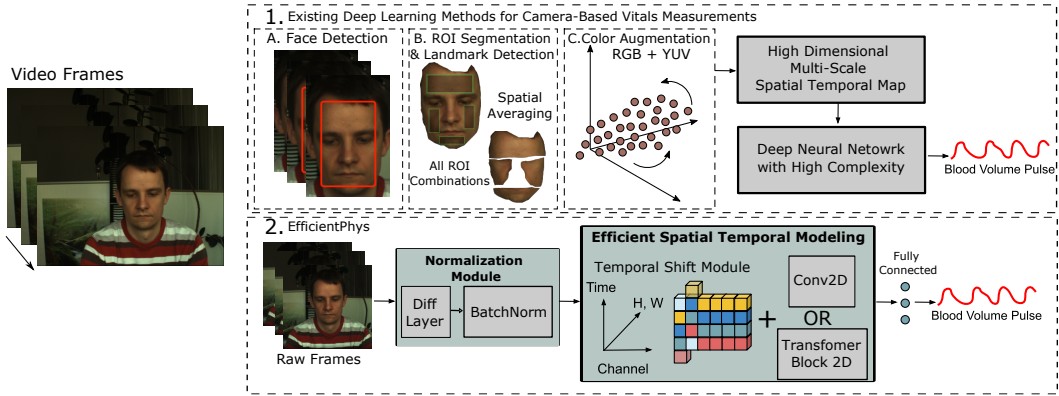

Figure 1: High-level comparison between EfficientPhys and existing deep learning approaches for camera-based vitals measurement

effort. Some libraries, such as facial landmark detection, are not even available on every platform. Thus, the last mile engineering using the existing methods becomes especially challenging.

Ideally, a video-based physiological measurement method would be able to run at a high frame rate even on mobile devices, be simple to implement across different platforms, and achieve state-of-the-art performance. Addressing the aforementioned challenges would help achieve these properties. We propose a truly end-to-end network, EfficientPhys, for which the input is unprocessed video frames without requiring accurate face cropping (see Fig. 1). Due to recent interest in visual transformers, we propose both a convolutional and visual transformer architecture and compare and contrast the performance of these two.

In summary, our key contributions are to: 1) propose two novel one-stop neural architectures, a visual transformer and a convolutional network, which do not require any preprocessing steps, 2) evaluate the proposed methods on three popular benchmark datasets, 3) evaluate on-device latency across both state-of-the-art machine learning based approaches as well as signal processing based techniques. To the best of our knowledge, this is the first paper that explores the visual transformer in camera-based physiological measurement and its comparison with convolutional networks. This is also the first paper exploring a completely end-to-end neural architecture. Our code and project page is available at here [1] and supplementary materials.

## 2 RELATED WORK

**Camera-based Vital Measurement.** There is a growing community studying the use of cameras to sense physiological vitals signs (Wu et al., 2000; Takano & Ohta, 2007; Verkruysse et al., 2008). Prior work established the fundamentals of how RGB images could be used to extract the pulse signal using signal source separation techniques (e.g., ICA) (Poh et al., 2010). Other methods derived these parameters from physically-based models to achieve elegant and fast demixing (e.g., Plane Orthogonal-to-Skin (POS))(Wang et al., 2017). By calculating a projection plane orthogonal to the skin-tone based on optical and physiological principles, the authors were able to achieve a stronger BVP signal-to-noise ratio (SNR).

Since the underlying relationship between the pulse and skin pixels is complex, deep convolutional neural networks have shown superior performance over the traditional source separation algorithms. DeepPhys (Chen & McDuff, 2018b) was the first paper that demonstrated that a deep neural network outperforms all the traditional signal processing approaches. Liu et al. have also proposed an on-device efficient neural architecture called MTTS-CAN for on camera-based physiological sensing, which leverages a tensor-shift module and 2D-convolutional operations to perform efficient spatial-temporal modeling (Liu et al., 2020b). More recently, an adversarial learning approach, called Dual-GAN, has also been studied to learn noise-resistant mappings from video frames to pulse waveform and noise distributions (Lu et al., 2021). With two generative-adversarial networks, they can promote

---

[1]https://sites.google.com/view/iclr22-efficientphys

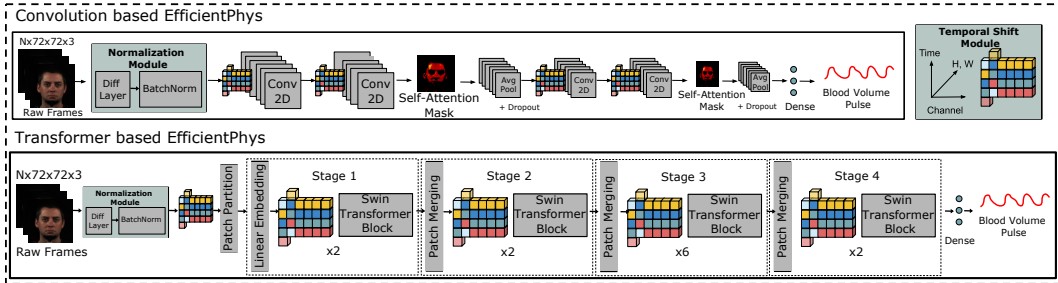

Figure 2: We present two novel architectures to enable simple, fast and accurate camera-based vitals measurement: Convolution based EfficientPhys and Transformer based EfficientPhys. N is the number frames of video clip inputting to the network.

each adversarial network's representation and further improve the feature disentanglement between pulse and various noise sources.

However, DeepPhys and MTTS-CAN both require a few preprocessing steps including calculating difference frames and performing image normalization. Dual-GAN has a even more complex preprocessing module called MSTMaps proposed by Niu et al. (2020). The MSTMaps is a multiscale spatial temporal map by 1) fine-grained facial cropping, 2) landmarker extraction, 3) performing average pooling for every color channel and every ROI combination for each frame, 4) generating ROI combinations using all the detected ROI regions and landmarkers, 5) multiplications of each item in all ROI combinations with six channel respectively. The final size of MSTMap is $(2^n - 1) \times T \times 6$ where $T$ is the number of frames and $n$ is the number of ROI regions. Such a preprocessing module not only takes large memory size but also could significant introduces large computational burden to the entire pipeline. Moreover, stacking all of these extra procedures makes development and deployment much more difficult. Unlike these previous work, our the goal of proposed EfficientPhys is to create a preprocessing-free neural architecture that is simple to use and deploy, efficient on mobile devices, and accurate on settings with various noises.

**Visual Transformers.** Although convolutional neural networks have been widely studied and used in many computer vision applications, vision transformer started showing its superior performance on the task image classification task. By training on larger datasets, vision transformer (ViT) attains excellent performance and can be used in downstream fine-tuning with fewer amount of data (Dosovitskiy et al., 2020). More recently, the state-of-the-art vision transformer, called Swin Transformer, was proposed to construct hierarchical feature maps and improve computational efficiency by using a hierarchical representation and limiting self-attention computation to non-overlapping local windows while allowing for cross-window connection (Liu et al., 2021b). However, transformer architectures have been merely studied in the field of camera-based vital measurement. The closest work is using transformer to detect remote photoplethysmography (rPPG) for for attack/spoofing detection (Yu et al., 2021). However, this paper did not have evaluation on the proposed vision transformer in the task of heart rate estimation using any public datasets, which is considered as the gold-standard benchmark for the field of camera-based vital measurement. To our best knowledge, our proposed vision transformer is the first architecture in camera-based heart rate measurement with detailed evaluation on various public datasets.

## 3 METHOD

### 3.1 CONVOLUTION BASED EFFICIENTPHYS

To enable simple, fast and accurate real-time on-device camera-based vitals measurement, we propose a one-stop-solution architecture that takes raw video frames as the input to the network and outputs a first-derivative PPG signal. The convolution based EfficientPhys is a one-branch network that contains a custom normalization layer, self-attention module, tensor-shift module and 2D convolution operation to perform efficient and accurate spatial-temporal modeling while making it simple to deploy.

**Normalization Module.** Existing neural methods all require different levels of preprocessing before providing the visual representation to the network to learn the underlying relationship between skin pixels and cardiac pulse signal. For instance, The state-of-the-art networks Dual-GAN (Lu et al., 2021) and CVD (Niu et al., 2020) proposed a hand-crafted spatial-temporal representations called STMaps. These preprocessed representations are generated for each video frame and includes steps of detecting 81 facial landmark points, extracting a set of region of interest (ROI) combinations ($2^n - 1$ where n is the number of ROIs, n=6) using these landmarks, and averaging pixel values in both the RGB and YUV color spaces, multiplying the 63 ROI combinations with the six channels. These modules not only add significant computational burden (Table 2 shows that Dual-GAN's preprocessing module takes 275ms per frame) but also make the system more challenging to implement and deploy on real-world computing systems such as mobile devices.

One of the goals of EfficientPhys is to remove these preprocessing modules entirely and provide a one-stop solution. To achieve such simplicity and deployability, we propose a custom normalization module, which can perform motion modeling between every two consecutive frames and normalization to reduce the lighting and motion noise. More specifically, the proposed normalization module includes a difference layer and a batchnorm layer. The difference layer (e.g., torch.diff) computes the first forward difference along the temporal axis of the raw video frames, by subtracting every two adjacent frames. To provide optical basis in our work, equation 1 illustrates the optical grounding of difference frame where $\boldsymbol{D}_k(t)$ of every two consecutive frames where $I(t)$ is the luminance intensity which is modulated by the specular reflection $\boldsymbol{v}_s(t)$ and the diffuse reflection $\boldsymbol{v}_d(t)$ as well as optical sensor's quantization noise $\boldsymbol{v}_n(t)$.

$$\boldsymbol{D}_k(t) = (I(t) \cdot (\boldsymbol{v}_s(t) + \boldsymbol{v}_d(t)) + \boldsymbol{v}_n(t)) - (I(t-1) \cdot (\boldsymbol{v}_s(t-1) + \boldsymbol{v}_d(t-1)) + \boldsymbol{v}_n(t-1)) \quad (1)$$

However, difference frames could be dramatically different in scale and make it hard for the network to learn meaningful feature representations, especially when the signal of interest is hidden in subtle pixel changes along the temporal axis and noise artifacts can cause significantly larger relative changes. To address this, we add a batch-normalization layer following by the difference layer. Adding a batchnorm layer provides two benefits: 1) it normalizes the difference frames to the same scale within the batch during training, 2) unlike fixed normalization in previous work (Chen & McDuff, 2018b; Liu et al., 2020a), batchnorm provides two learnable parameters $\beta$ and $\gamma$ for scaling (to a different variance) and shifting (to a different mean) and two constant parameters which are the mean $\mu$ and the standard deviation $\sigma$. Through the learning process, the batchnorm layer can learn the best parameters for removing noise as the Equation 2 shows. Without a batchnorm layer, directly applying a difference layer means the frames appear "black"; because the subtle changes of skin pixels in every two consecutive frames are relatively very small. On the other hand, adding a follow-up batchnorm layer will help it learn the normalization function to magnify the subtle changes of skin pixels substantially. The result is not simply a magnification of values but a normalization and magnification. Moreover, we also compare the output batchnorm layer to the hand-crafted normalized frame as shown in Fig.3. The output of batchnorm layer contains more information and qualitative analysis suggests it should be a better tool for skin segmentation after the learning process.

$$\boldsymbol{N}_k(t) = \frac{(\beta_t * \boldsymbol{D}_k(t) + \gamma_t) - \mu_{\boldsymbol{D}_k}}{\sigma_{\boldsymbol{D}_k}} \quad (2)$$

**Self-Attention-Shifted Network.** To efficiently capture the rich spatial-temporal information, we propose a self-attention-shifted network (SASN). SASN is built on top of the previous state-of-the-art method for on-device spatial-temporal modeling in optical cardiac measurement - tensor-shift convolutional attention network (TS-CAN) (Liu et al., 2020b). TS-CAN has two convolutional branches, one of which takes a preprocessed difference frame representation and one of which takes a normalized appearance frame. The motion branch performs the main spatial-temporal modeling and estimation, and the appearance branch provides attention masks to guide the motion branch to better isolate the pixels of interest (e.g., skin pixels). However, we argue that the attention masks do not have to be obtained through a separate appearance branch and they can be also learned with a single branch end-to-end network. As Fig. 2 illustrates, our proposed self-attention-shifted network starts with the custom normalization module discussed in the previous section then continues with two tensor-shifted convolutional operations. After the second and fourth tensor-shifted 2D convolutional layers, we add

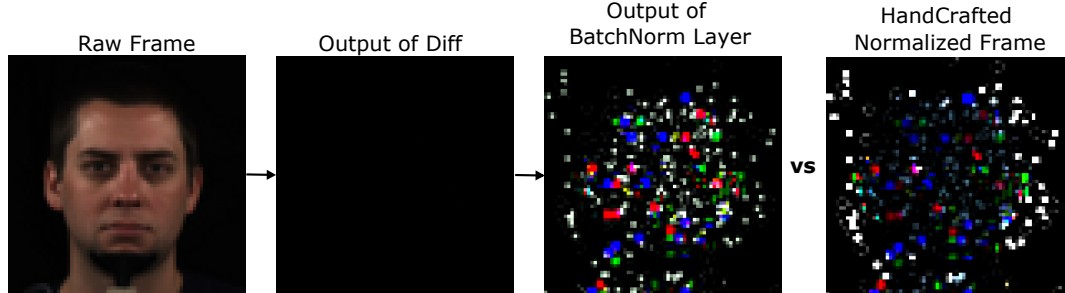

Figure 3: Outputs of Diff and batchnorm layers and comparison with normalized frames generated via the hand-crafted process in prior work Chen & McDuff (2018b). The output from the diff layer is almost black because the difference in skin pixels of consecutive frames is very subtle.

a self-attention module respectively to help the network minimize the negative effects introduced by tensor shifting as well as motion and lighting noises. The self-attention layers are softmax attention layers with 1D convolutions followed by a sigmoid activation function. Then, normalization is applied to remove the outlying values in the attention mask, and the final normalized attention mask is element-wise multiplied with the output from the tensor-shifted convolution. Equation 3 summarize how our self-attention mechanism works where $ts(.)$ denotes tensor shift operation, $\omega_c^t$ denotes the 2D convolutional kernel followed by the tensor shift module, and $\omega_a^t$ is the $1 \times 1$ convolutional kernel for self attention.

$$(\omega_c^t ts(\boldsymbol{N}_k(t)) + b_c^t) \odot \frac{H_t W_t \cdot \sigma(\omega_a^t \mathbb{X}_\alpha^t + b_a^t)}{2 \parallel \sigma(\omega_a^t \mathbb{X}_\alpha^t + b_a^t) \parallel_1} \tag{3}$$

## 3.2 TRANSFORMER BASED EFFICIENTPHYS

**Efficient Spatial-Temporal Video Transformer.** Due to the recent success of visual transformers for image and video understanding and the importance of attention mechanisms for this task (Chen & McDuff, 2018a; Yu et al., 2019; Qi et al., 2020; Liu et al., 2020b), we also present a visual transformer version of EfficientPhys. For this task, we need a visual transformer to learn both spatial and temporal representations. Several existing video-based visual transformers are based on 3D-embedding tokens and input all the frames into 3D encoder and spatial-temporal attention modules (Arnab et al., 2021; Liu et al., 2021d). However, the computational complexity makes that unfavourable for real-time efficient modeling on mobile devices. In the convolutional version we used tensor-shifted 2D convolutions which have been shown to achieve comparable performance as 3D convolutions (Liu et al., 2020b). Inspired by this, our proposed transformer based EfficientPhys is based on a 2D visual transformer, Swin transformer (Liu et al., 2021c), but with added components that we will describe below.

Since the 2D Swin transformer is only able to learn spatial features that map raw RGB values to latent representations between a single frame and the target signal (pulse) and does not have ability to model temporal relationships beyond consecutive frames. One of the main contributions of the Swin transformer is the shifted window module which has linear computation complexity and allows cross-window connection by shifting the window partition and limiting self-attention computation to non-overlapping local windows. Inspired by the idea of shifting of spatial window partitions, we propose to add a tensor-shift module (TSM) (Lin et al., 2019) before every Swin transformer block to facilitate information exchange across the temporal axis. The TSM first splits the input tensor into three chunks, shifts the first chunk to the left by one place (advancing time by one frame) and shifts the second chunk to right by one place (delaying time by one frame). All the shifting operations are along temporal axis and performed before the tensor is fed into each transformer block as shown in Fig. 2. By adding the TSM module to the Swin transformer, the new transformer architecture now has the ability to perform efficient spatial-temporal modeling and attention by combining shifting window partitions spatially and shifting frames temporally. It is worth noting that TSM does not introduce any learnable parameters thus the proposed transformer architecture has the same number of parameters as the original Swin transformer. Finally, to enable truly end-to-end inference and

learning, we also add the same normalization module proposed in the convolution EfficientPhys to this architecture.

In summary, the transformer-based EfficientPhys is the first end-to-end transformer architecture for camera-based cardiac pulse measurement that leverages tensor-shift modules and window-partition shift modules to perform efficient spatial-temporal modeling and attention to learn the underlying physiological signal from skin pixels.

## 4 EXPERIMENTS

**Training Data.** To help create a robust and generalizable model for cross-dataset evaluation we use two datasets. The first is AFRL (Estepp et al., 2014), which includes 300 videos from 25 subjects (17 males and 8 females). For each video, the raw resolution is 658x492 and the sampling rate of the synchronized pulse measurement is 30Hz. The dataset includes videos with a range of head motions. Every participant was instructed to maintain stationary for the first two tasks, and then to perform head motions with increasing rotational velocity in the next four tasks (turning from left to right). Along with AFRL dataset, we also leverage a synthetic avatar video dataset introduced by McDuff et al. (2020) where each synthetic video is parameterized and generated with a custom pulse signal, background, facial appearance, and motion. More specifically, the input pulse signal is used to augment skin color and the subsurface radius of skin pixels to mimic the effect of the blood volume pulse on the skin's appearance. Synthetic data such as this introduces greater diversity into the training set and has been shown to effectively help reduce disparities in performance by skin type.

**Testing Data.** We use three popular benchmark datasets to evaluate the accuracy of the proposed EfficientPhys. UBFC (Bobbia et al., 2019) is a dataset of 42 videos from 42 subjects, and the raw resolution of each video is 640x480 in a uncompressed 8-bit RGB format. The sampling for synchronized pulse signal is 30 Hz. All of the tasks collected in UBFC are stationary. MMSE (Zhang et al., 2016) is a dataset including 102 videos from 40 subjects, and the raw resolution of each video is at 1040x1392. The ground-truth waveform for MMSE is blood pressure signal instead of blood-volume pulse signal, and the sampling rate is 25hz. It is worth noting that MMSE contains a diverse distribution of skin types in Fitzpatrick scale (II=8, III=11, IV=17, V+VI=4). PURE (Stricker et al., 2014) is a containing 60 videos from 10 subjects. The raw resolution of each video is 640x480, and the sampling rate of ground-truth pulse signal is 60 Hz. PURE includes a diverse set of motion tasks such as steady, talking, slow/fast translation between head movements and the camera plane, small/medium head rotation.

**Implementation & Experiment Details.** We implemented both convolution based and transformer based EfficientPhys in PyTorch (Paszke et al., 2019). We used an AdamW optimizer to train both networks instead of Adam by introducing additional regularization to reduce the effects of over-fitting through weight decay (Loshchilov & Hutter, 2017). The learning rate we used for Convolutional model was 0.001 while the one for transformer model was 0.0001. Based on empirical studies, we used mean squared error(MSE) loss for training the transformer models and negative pearson loss (Tsou et al., 2020) for the convolutional model. We trained both models for five epochs. We implemented TS-CAN based on the open-sourced code (Liu et al., 2021a; 2020b). To calculate the performance metrics, we first applied a band-pass filter to the signal with a cutoff frequency of 0.75 and 2.5Hz (45 beats/minute to 150 beats/minute). We then followed Dual-GAN's evaluation scheme to run peak detection and FFT to get estimated heart rate on each video of UBFC and PURE datasets (Lu et al.) and MetaPhys's evaluation schema on MMSE (Liu et al., 2021a). We calculated three standard metrics for each video: mean absolute error (MAE), root mean squared error (RMSE) and Pearson correlation ($\rho$) in heart rate estimations and the corresponding ground-truth heart rates from the blood volume pulse collected via contact oximeter sensor.

To explore the efficiency of different architectures on mobile devices, we also conducted experiments on a quad-core Cortex-A72 Raspberry Pi 4B to evaluate the model's performance on an edge device. We performed inference 10 times to get a reliable averaged on-device inference latency for EfficientPhys and TS-CAN. Due to the lack of open-source implementation from Dual-GAN, we were only able to find the implementation of STMaps which is the preprocessing module of Dual-GAN. Thus, we only evaluated the on-device latency for the preprocessing module in Dual-GAN. We also evaluated the latency of POS, CHROM, and ICA as they are traditional signal processing methods and don't have a separate preprocessing module.

Table 1: Cross-dataset heart rate evaluation on three public datasets: UBFC, MMSE and PURE (beats per minute)

| | UBFC | | | PURE | | | MMSE | | |
|---|---|---|---|---|---|---|---|---|---|
| Method | MAE↓ | RMSE↓ | $\rho$↑ | MAE↓ | RMSE↓ | $\rho$↑ | MAE↓ | RMSE↓ | $\rho$↑ |
| EfficientPhys-C | **1.14** | **1.81** | **0.99** | **1.33** | **5.99** | **0.97** | **2.91** | **5.43** | **0.92** |
| EfficientPhys-T1 | 1.53 | 2.27 | 0.99 | 2.15 | 7.98 | 0.94 | 3.48 | 7.21 | 0.86 |
| EfficientPhys-T2 | 3.07 | 4.78 | 0.96 | 7.71 | 9.95 | 0.85 | 3.51 | 6.98 | 0.88 |
| TS-CAN(Liu et al., 2020b) | 1.70 | 2.72 | 0.99 | 2.48 | 9.01 | 0.92 | 3.85 | 7.21 | 0.86 |
| Dual-GAN(Lu et al., 2021) | **0.74** | **1.02** | **0.99** | N/A | N/A | N/A | N/A | N/A | N/A |
| PulseGAN(Song et al., 2021) | 2.09 | 4.42 | 0.99 | N/A | N/A | N/A | N/A | N/A | N/A |
| POS(Wang et al., 2017) | 3.52 | 8.38 | 0.90 | 3.14 | 10.57 | 0.95 | 3.98 | 6.66 | 0.67 |
| CHROM(De Haan & Jeanne, 2013) | 3.10 | 6.84 | 0.93 | 2.07 | 9.92 | **0.99** | 7.31 | 9.85 | 0.57 |
| ICA(Poh et al., 2010) | 4.39 | 11.60 | 0.82 | N/A | N/A | N/A | 10.2 | 14.4 | 0.50 |

MAE = Mean Absolute Error in HR estimation, RMSE = Root Mean Square Error in HR estimation, $\rho$ = Pearson Correlation in HR estimation.

Table 2: On-Device Data Preprocessing Latency and Model Inference Latency Per Frame (ms)

| Method | Preprocessing (ms) ↓ | Model (ms) ↓ | Total (ms) ↓ |
|---|---|---|---|
| EfficientPhys-C | 0 | 40 | 40 |
| EfficientPhys-T1 | 0 | 300 | 300 |
| EfficientPhys-T2 | 0 | 40 | 40 |
| TS-CAN(Liu et al., 2020b) | 3 | 60 | 63 |
| Dual-GAN(Lu et al., 2021) | **275** | N/A | > 275 |
| POS(Wang et al., 2017) | 0 | 27 | 27 |
| CHROM(De Haan & Jeanne, 2013) | 0 | 28 | 28 |
| ICA(Poh et al., 2010) | 0 | 31 | 31 |

ms = Preprocessing and model latency on Raspberry Pi 4B per frame.

## 5 RESULTS AND DISCUSSION

**EfficientPhys vs. State-of-the-Art.** In the Table 1, we present results from our proposed Efficient-Phys models and the current state-of-the-art neural and signal processing methods. The learning models are all trained on the same datasets and tested on a separate dataset (e.g., UBFC) to test if the model can be generalize to videos with a different facial appearance, background and lighting. To investigate how the depth of the network impacts the Transformer architecture, we created two version of Transformer-based EfficientPhys: T1 and T2. T1 uses the same depth as the Swin Transformer reported in Liu et al. (2021c) ([2, 2, 6, 2]). Each number indicates the number of Swin Transformer blocks as illustrated in Fig. 2. T2 has a much lighter architecture to enable real-time on-device inference which has a depth of [2, 1]. EfficientPhys-C denotes the Convolution-based EfficientPhys as shown in the Fig.2. EfficientPhys-C and EfficientPhys-T1 outperform almost all the existing method except Dual-GAN. However, we argue that the margin is relatively small as both Dual-GAN and EfficientPhys-C achieve a Pearson correlation of 0.99. EfficientPhys-C again surpasses all the state-of-the-art methods by 46% of MAE in PURE and 26% of MAE in MMSE. Unfortunately, Dual-GAN did not perform cross-dataset evaluation on other datasets. Due to the lack of open source implementation or released models, we could not successfully replicate their complicated model architecture.

**Computational Cost and On-Device Latency.** Fig. 4 and the Table 2 summarize the computational cost of the existing neural methods. Again, due to the lack of open source implementation and complex algorithm design, we were not able to replicate every architecture to benchmark its on-device latency. The results show that EfficientPhys-C only takes 40ms to process a single frame and it does not take any extra computational time to perform preprocessing. On the other hand, due to the complexity model architecture and extra time for calculating hand-crafted normalized raw and difference frames, TS-CAN takes 63ms per frame. As mentioned earlier, Dual-GAN has a complicated preprocessing procedure for facial landmark detection, segmentation, color transformation and augmentation. We implemented this and benchmark the preprocessing module on our platform, and it takes 275ms per frame, which is already 7x than the entire computational time of EfficientPhys-C. The estimation

network in Dual-GAN also includes 12 2D convolution operations, numerous 1D convolution operations. Thus, we believe it would add a significant amount of computational time on top of the 275ms preprocessing time per frame. The default Transformer-based EfficientPhys (T1) has a unfavorable inference due to its deep architecture design and takes 300ms to process every single frame. After reducing the depth to EfficientPhys-T2, it can achieve same inference time as the EfficientPhys-C. However, EfficientPhys-T2 has a much worse performance on all three benchmark datasets.

**Convolution vs. Transformer in Camera-Based Vitals Measurement.** Although visual transformers have begun to achieve state-of-the-art performance on some vision tasks, it is not the case in the task of video based vitals measurement. Based on the results shown in Table 1, Efficient-C outperforms both Efficient-T1 by 25% of MAE in UBFC, 38% of MAE in PURE, 16% of MAE in MMSE, while Efficient-C is more than 7x faster in terms of latency. When we shrink the Transformer-based EfficientPhys to a similar complexity as Convolution-based EfficientPhys, the accuracy performance is significantly diminished. The errors from lightweight Transformer-based EfficientPhys-T2 increased 106% of MAE in UBFC, 478% of MAE in PURE and 21% of MAE in MMSE. These results indicate a shallow transformer architecture struggles to model subtle changes of skin pixels in the video. These finding suggest two potential insights. First, further optimizations will be necessary for transformers to outperform, even relatively shallow, convolutional models in this domain, this is possibly especially true when there is not a large amount of high-quality data available. As previous studies have shown

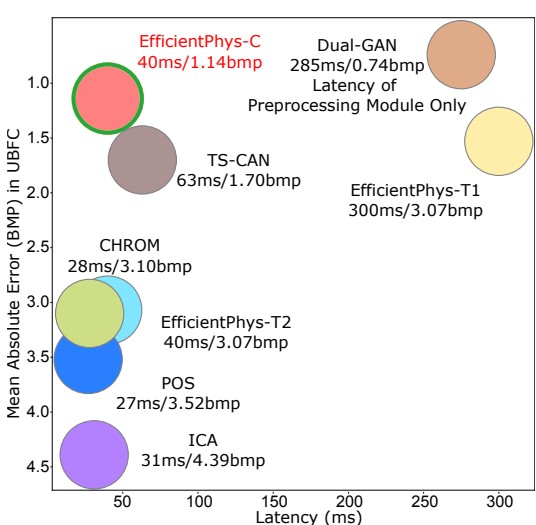

Figure 4: Accuracy-Latency Trade-off in eight different methods. Y-axis denotes the MAE error, and X-axis denotes the latency. The methods in the left-top corner have the best accuracy-latency Trade-off.

(Dosovitskiy et al., 2020), Transformers usually require more pre-training samples to obtain state-of-the-art accuracy. Unfortunately, currently the amount of data in the field of camera-based vital measurement is limited compared to other visual tasks. We believe synthetic data is one way to help address this issue. Second, the good accuracy-efficiency trade-off for visual transformer might not be scaled to on-device architectures without further work. Since many on-device neural networks require significantly less amount of computing resources to perform real-time operations, scaling the Transformer architecture down is not ideal as our experimental results of EfficientPhys-T2 have shown.

**Simplifying Last-Mile ML Deployment.** Numerous real-world applications are driven by novel machine learning algorithms. However, deploying these algorithms on different computing platforms has been extremely challenging for various reasons. One of these is researchers sometimes only pay attention to the accuracy of the model and ignore the complexity of the last-mile engineering efforts. In this paper, we address this important issue through our one-stop architecture which takes the unprocessed raw frames and directly outputs the desired signal. This elegant and simple design not only will reduce the burden of engineering required for cross-platform implementations, but also will help the research community to replicate and reproduce results.

**Extensible to Other Signal.** Finally, another potential upside of our end-to-end design and the low latency, we envision EfficientPhys could be applied to various other video-based applications. Since the input of our model is raw frame, we believe EfficientPhys can be easily extended to other physiological signals and formats of data such as video-based blood pressure measurement and video understanding & recognition etc. On the other hand, most of the baseline methods we compared (e.g., Dual-GAN, PulseGAN) require many custom preprocessing operations for video-based measurement which are less useful in other applications.

## 6 BROADER IMPACTS AND ETHICS STATEMENT

Recent Information and Communications Technologies for Development (ICT4D) work reconceptualize technology development to empower the people it serves. Machine learning based applications such as health interventions also tend to focus more on the development of the technology itself rather than the people and problems they address. Although many large-scale deep neural network models are trained on the data created by the public, they are often not freely available to the public. In this work we only used data that was collected under informed consent for the purposes of physiological analysis. We also make the trained models accessible and available to more diverse communities as described below.

During the development of EfficientPhys, we intended that the innovations do not create larger disparities between different populations. By achieving the state-of-the-art accuracy and efficiency as well as our simple and elegant design, we believe EfficientPhys will help make camera-based vitals measurement more widely available to the medical research community and broader community in computing. We also believe that this technology can have a particular impact in low-resource settings there are greater barriers to access healthcare. We envision our proposed method could, with the appropriate clinical validation and regulatory approval, eventually be used in healthcare applications (e.g., real-time vitals measurement in telehealth appointments). During the COVID-19 pandemic the need for such technology has been clearly highlighted. We contextualize our contributions within the scope of democratizing technology for social good and helping to reduce health disparities with advanced AI technology. However, we are aware that machine learning systems are biased and can propagate inequalities. Before technology such as that presented in this paper is ready for deployment we need to make sure that that is not the case.

## 7 REPRODUCIBILITY STATEMENT

We described detailed experimental details in the section of 4. We also release our code in a Github anonymous repo and provide a detailed project overview in an anonymous website: `https://sites.google.com/view/iclr22-efficientphys`.

## 8 CONCLUSION

In this paper, we present a novel method called EfficientPhys to enable simple, fast, accurate camera-based contactless vitals measuremnt. We achieved the state-of-the-art performance with using significant less computational power. With the simple and elegant one-stop design, EfficientPhys also help address the issue of last-time machine learning deployment and reduce health disparity.

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

## A  APPENDIX

Project and Code Link: `https://sites.google.com/view/iclr22-efficientphys`

