# OpenReview forum: "EfficientPhys: Enabling Simple, Fast, and Accurate Camera-Based Vitals Measurement"
_ICLR.cc/2022/Conference — ICLR 2022 Submitted_

### Official Review · Reviewer_S4mf · 2021-10-29

**Correctness:** 3
**Technical Novelty And Significance:** 3
**Empirical Novelty And Significance:** 3
**Recommendation:** 8
**Confidence:** 4

**Main Review:**

Strengths:
- The novelty of the approach resides in its aim to efficiency, mainly obtained by removing the preprocessing phase. The specific convolutional version leverages a spatial-temporal attention based on a parameter-less tensor shift, with a further gain in efficiency.
- The outline of the paper is clear and it is generally well written, with some minor exception. Both the convolution and transformer approaches are described in detail.
- The comparison shows that convolutional version of EfficientPhys outperforms other state-of-the-art approaches. Interestingly, it also shows that the this variant is more efficient than the transformer-based one since it obtains a greater accuracy if the models are constrained to the same size. It would be interesting to explore how much general is this assessment about the comparison of the two architecture.
- references seem accurate, comprehensive and up to date.
- code is released as public github repository.

Weaknesses:
- The choice to call with the same name (EfficientPhys) two different approaches is a bit confusing. Models are indeed three: EfficientPhys-C (convolution), EfficientPhys-T1 (transformer-regular), EfficientPhys-T2 (transformer-shrinked). They share the same normalization block and arguably a similar output block, but with a different core. They also perform in a very different way: only '-C' outperforms other approaches in terms of MAE, RMSE, ro (except DualGAN), and is the Pareto optimum for MAE/latency; '-T1' is good in terms of MAE, RMSE, ro, but not in latency; '-T2' in contrast show interesting latency but not so good accuracy. It is difficult to consider these models as unique, and results can not be claimed for EfficientPhys but for EfficientPhys-C.
- the lack of a complete comparison with the best other approach, DualGAN, is disappointing. As explained by the authors, this is due to the lack of results of DualGAN on the public datasets PURE, MMSE, and cannot be totally addressed to them.
Nonetheless, this diminishes the claimed results since DualGAN outperforms EfficientPhys models in UBFC dataset.
- Self-Attention-Shifted Network is described by eq. 3 which is too verbose and somehow obscure, and needs to be better explained as it describes a core component of the model. It fails to give an intution of how the module works.
- It is not explained at all which is the final task (and consequently modules on top of architecture) of both models: probably regression?
- Fig. 2: the normalization module seems different in the two versions, but reading the text it seems the very same. Figures are great value for giving an intuition of how a system works, but a standardization of the pictograms is needed.
- Fig. 4 is a bit confusing in the 0/50 latency range, 2.5/4.0 MAE: the chosen symbols overlap.
- minor problems about the text:
-- pag. 4, after eq. 1: 'To address this, we add a batch-normalization layer followed by the difference layer' should be: 'To address this, we add a batch-normalization layer following the difference layer'
-- pag. 5, second paragraph of section 3.2: the first sentence, 'Since the 2D Swin transfromer...', is not correct and needs to be rephrased
-- pag. 9, last two sentences of section 6: 'However, we are aware...': they are quite involuted and the meaning is not clear.


**Summary Of The Paper:**

The paper introduces two novel neural models to address the task of camera-based physiological measurement: the first model leverages convolutional layers and a spatio-temporal attention module; the second model is based on Vision Transformer.
They are described as variants of the same network called EfficientPhys, and both remove the need of specific preprocessing steps on input data by introducing a normalization module.
The resulting lightweight architecture makes both variants easy to be implemented in multiple devices such as mobiles, and to be applied to the elaboration of different physiological signals.
EfficientPhys is tested on three public datasets with state-of-the-art approaches obtaining good results in both accuracy and latency.


**Summary Of The Review:**

The paper is an interesting contribution in the task of camera-based physiological measurement.
Two neural models are presented with two different approaches: convolution, and vision transformer.
They are alternative version of the same network called EfficientPhys, and represent valuable examples of efficient deep learning: they show a lightweight pipeline having removed the data preprocessing, and a spatio-temporal attention build on the parameterless tensor shift (for the convolution version). Nonetheless, they obtain good performance in terms of accuracy (MAE, RMSE, ro) and latency, with the convolutional model in the Pareto frontier.
The inherent lightness of the architecture makes the models portable to multiple devices platforms, facilitating the access to healthcare in low resources scenarios.
For these reasons the paper is to be accepted, with minor changes mainly to improve the readability.

---

### Official Review · Reviewer_kqwX · 2021-11-02

**Correctness:** 3
**Technical Novelty And Significance:** 2
**Empirical Novelty And Significance:** 2
**Recommendation:** 3
**Confidence:** 4

**Main Review:**

Let me summarize the strengths of the paper first.
+ The presented approach is simple and easy to implement.
+ There is an exploration of different designs of the backbone networks.
+ The performance of the presented method is strong.

The weaknesses of the paper are as follows.
- The writing of the paper needs improvements. There are many grammatical errors. For example, in the abstract, "state-the-art-performance" and "Prior research have" -> "Prior research has." The authors are recommended to have a few revisions on the paper.
- I do not think the presented architectures are technically novel. They are built with existing backbones. Also having an end-to-end network in deep learning applications is straightforward.
- There are no ablation studies on different designs in the presented model.
- The results on UBFC by the proposed method are not very competitive.
- I do not get the point of presenting a model with transformers. This model apparently performs worse than the one with convolutions in accuracy and speed.

**Summary Of The Paper:**

In this work, the authors present simple and efficient models for camera-based vital measurement. The proposed approach does not rely on any pre-processing steps but uses end-to-end neural networks. The authors present two types of neural networks, one based on convolutional neural networks and one based on transformers. The experiments show the approach archives state-of-the-art performance on three public datasets with fast inference speed.

**Summary Of The Review:**

In summary, I feel the work lacks sufficient technical contribution in terms of novelty and empirical motivation. The experimental results are good but somehow expected. The writing of the paper needs improvements.

---

### Official Review · Reviewer_B8NS · 2021-11-02

**Correctness:** 2
**Technical Novelty And Significance:** 2
**Empirical Novelty And Significance:** 2
**Recommendation:** 3
**Confidence:** 4

**Main Review:**

The paper shows superior results on multiple public datasets that are moderately difficult. One dataset include darker skin tones which are known to be more difficult for rPPG, and another dataset includes some head motion. The algorithm proposed are fast and can be deployed on mobile platform, which will help improve access to the technology in general, even though, for the topic at hand, accessibility is not the main focus here.

The machine learning aspect of this paper seems solid. The training and testing sets are clearly separated. All the basic information about the datasets that are important to understanding how ML is applied is clearly laid out in the experiment section.

My main concerns for this paper is around evaluation. Several aspects of evaluation needs to be improved:
  - The test datasets are all relatively small. The biggest dataset only have 100 videos. Furthermore, the diversity and difficulty of the datasets are unclear. UBFC contains stationary videos, MMSE has only 4 subjects of the darkest skin types, and PURE only have at most moderate head motion. I recommend the author to perform evaluation on larger, more challenging datasets such as MAHNOB-HCI, and VIPL-HR, which contains many more videos (2k in the latter, and several hundered in the former).
  - Some of the comparison in Table 1 is misleading, and I would love some additional clarification from the authors. The result for Dual-GAN seems to be copied from Table 5 of the original paper, where the model is trained on PURE, and test on UBFC. Since the training data is different from the current method (AFRL), the comparison is not direct. It's possible that Dual-GAN will do even better if trained on AFRL, or the other way around, where the proposed method could be even better if trained on PURE. I recommend the author to actually retrain comparisons on the exact same dataset, so that we can be sure that the proposed method actually improve on the state of the art.

  In this particular case, Dual-GAN actually have better numbers than the proposed method. I would like to get some clarification on how the rest of the comparison is done. This concerns also apply to the efficiency evaluation. If the competing method have not been properly optimize, its runtime can suffer significantly. How has the authors optimized competing method's implementation onto their Raspberry Pi platform?

Secondary to evaluation, it is unclear how swin transformer and tensor shifting actually help authors avoid preprocessing steps required by other methods. These are the components that have been used in previous work, and they still require some preprocessing in order to achieve good performance. One possible explanation is the choice of testing dataset that are relatively easy and contain limited amount of motion. I would love to see a more apple-to-apple comparison with previous methods (using the same dataset, well-tuned model) before I am convinced that the proposed method actually works.

Relatedly, the authors claim to be the first end-to-end network, but I could find their custom normalization and diffing to be a sort of preprocessing, in the same spirit as some of the previous work (DeepPhys by Chen and McDuff).

Lastly, the method section contains many unjustified technical decision, and also some inaccurate statements. I would love to see the author's clarification on these:
Page 4:
“which can perform motion modeling between every two consecutive frames and normalization to reduce the lighting and motion noise.” – This claim that the normalization module is able to reduce lighting and motion noise hasn’t been shown in this paper.

Equation 1: camera noise has more components than just quantization. Shot noise, read noise, are two other important sources of noise that was not considered here. There are also processing functions (sensor response can be non-linear), that were not discussed or taken into consideration. While these could be approximated away in most cases, the authors need to be precise about their treatment of these effects.

“the batchnorm layer can learn the best parameters for removing noise as the Equation 2 shows.” Assuming that noise between sample is IID, why is it desirable to normalize across batch?

last paragraph: “we argue that the attention masks do not have to be obtained through a separate appearance branch and they can be also learned with a single branch end-to-end network” What is the author’s justification for this?

Page 5:
Figure 3: It’s unclear what difference color pattern means. These could easily be an artifact of color scale used in plotting. Also, if the concern is the difference frame will be close to zero, wouldn’t normalize it to zero mean, unit SD be enough?

Additional suggestions:

Additional results by groups could shed more light on what are the weaknesses and strengths of the proposed method. For example, how does the model perform in each sky type group, each motion amount group, etc.


**Summary Of The Paper:**

This paper propose an end-to-end learning-based method to extract heart rate from facial videos. The authors proposed two networks--a convolutional based, and transformer-based. Both network utilize tensor shifting to encode temporal dimension. The Conv-based one utilize self-attention to guide the network towards important region of the image, while the transformer-based one uses swin transformers. The authors implemented their network in rasberry pi and show that their method is more accurate than the comparison on three relatively small public datasets, while being 33% faster in processing time due to lack of preprocessing.

**Summary Of The Review:**

At face value, this paper produce superior HR detection result, and more efficient algorithm. However, at closer inspection, a more direct comparison, and evaluation on a more challenging dataset are still needed to validate the effectiveness of the algorithm. In my opinion, the contribution is still not enough for a top-tier venue like ICLR.

---

### Official Review · Reviewer_anoh · 2021-11-03

**Correctness:** 3
**Technical Novelty And Significance:** 3
**Empirical Novelty And Significance:** 2
**Recommendation:** 5
**Confidence:** 4

**Main Review:**

# Strengths

1.  The application is important and it will certainly be easier to deploy models which have fewer requirements around input data preprocessing - so I think the motivation for the paper is good and I appreciate the emphasis on trying to make these systems useful in the real world.
2. I believe this is the first successful application of a transformer architecture for remote ppg estimation. The approach seems clean. The introduction of a tensor-shift module before Swin transformer blocks seems to be a neat way to get some temporal propagation of information. The CNN based model also appears to be simpler than the previous work it builds on, by removing the need for an input appearance branch and associated compute by virtue of adding a self attention layer. It is interesting to me that using batch-normalized difference frames at input appears to be sufficient for extraction of the ppg signal.
3. The cross-dataset experiment is a good way to evaluate generalization, because different datasets are known to exhibit different kinds of biases.
4. Releasing the code will be beneficial to others in the community.

# Weaknesses

1. The estimation task in the paper is concerned with photoplethsymography/heart rate. Although the authors hint to the use of the model for other estimation tasks (e.g. blood pressure) in future, it is not actually assessed. Since "vitals" and "physiological measurement" encompass more than heart rate estimation (e.g. respiratory rate, blood pressure, temperature), I believe this work should be described as a technique for remote PPG / BVP / heart rate monitoring (and not the more generic "vitals" or "physiological measurement") unless experiments can be added which demonstrate otherwise. To avoid misleading readers, I think this ought to be made clearer in both the title and the abstract.

2. The paper clarity could be improved in places and sometimes appears to lift from the work of MTTS-CAN while losing a few important details, e.g.
- it is unclear what the value of Eq 1 is to understand this work - none of these terms are ever mentioned again. Perhaps the motivation underlying this equation could be better explained through simple English or a figure?
- Figure 2 does not label the dense layer; "volumne" typo; Temporal Shift Module should ideally be referenced? Is the self-attention mask illustrated here an actual output from the EfficienctPhys-C model?
- Fig 4 which is similar to a figure from the MTTS-CAN paper but here the circle radius is unused? Could the circles be resized? Y axis should be BPM?
- Eqn. 3 is not very clear to understand, without being familiar with the “Attention on Temporal Shift” section of the TS-CAN paper. Please add further explanation with reference to the past work to explain how it improves. Please also choose a different term than `ts` to denote tensor-shift, as the `t` of `ts` may be confused with time.

3. While I agree that pre-processing can make deployment complicated, I don't think this issue has been sufficiently explored to be able to call this a technical claim of the paper. It should also be pointed out that certain preprocessing steps such as face tracking are increasingly readily available, even on lightweight devices. The fact that the proposed model doesn't require face tracking is likely to be because it is (1) implicitly doing it, thanks to the training data and/or (2) makes assumptions such as the face being roughly centered, taking up a certain proportion of the image, and the background remaining static throughout. I think the paper would benefit from pointing out or exploring these things. Taken to the extreme, if the input data is 10MPx, and the subject's face takes up 0.1 MPx, then a lightweight face detection module could potentially save a lot of compute/memory that these proposed methods would not -- so even though this model does not require preprocessing, it does rely on assumptions about how the input data is captured, which should be clarified.

4. At low error rates on these three test datasets, errors caused by one or two subjects or video clips can make a big difference between "SOTA" or not. The datasets themselves are also known to contain erratic ground truth. Was any filtering carried out to remove erroneous samples from the evaluation? Do EfficientPhys-C and EfficientPhys-T1 tend to make similar errors on the test datasets?

5. I understand from Table 2 that  “The learning models are all trained on the same datasets”, but this probably only applies to the methods which have all columns populated? For example, Dual-GAN’s numbers for UBFC are taken from Table 5 of their paper, which is based on training on PURE and testing on UBFC. In addition Dual-GAN reports rho as 0.997 which has been rounded to 0.99 here. I think it should be made clear that this is not a fair comparison, and it is unclear which method actually benefits from this unfairness. It is no fault of the authors that Dual-GAN’s method is hard to reproduce, but it would be good to be more transparent here even if it makes the narrative a bit murkier.

6. The paper highlights the benefits of batch normalization, but other than a qualitative example in Fig 3, does not show any experiment results to highlight its benefits. Moreover, the benefits of batch normalization may depend on the batch size used in training, but I could not find any details about this in the paper. Please could these details be added to the paper?

7. TS-CAN seems to be the main benchmark for this work. I understand EfficientPhys-C differs in several ways (removal of appearance branch, additional of bnorm, addition of self-attention); it would be valuable to see the effects of these various changes - do they rely on each other to get the performance benefit?

8. I understand EfficientPhys-T1 and T2 differ by the number of blocks, [2,2,6,2] -> [2,1]. Were any other alterations attempted to reduce the compute requirements, such as varying patch sizes, or number of heads, etc?

9. Reproducibility: it is great that training code for this model can be shared and that the evaluation is on public datasets. However, is the synthetic dataset available for other researchers to be able to reproduce this work? If the synthetic data cannot be shared prior to the conference, the paper would benefit by providing alternative reproducible experiments - e.g. leave-one-out cross dataset training on public datasets, otherwise it will not be possible for others to build on this work.

10. More characteristics about the synthetic data should be recapped in Section 4, as is done for the other datasets (e.g. resolution used, diversity etc.). What proportion of the final training dataset was real vs synthetic? Were any experiments undertaken to measure the variation of EfficientPhys models’ performances vs amount of synthetic data (the paper mentions “We believe synthetic data is one way to help address this issue [of limited availability of high-quality training data]” - was this measured in practice)?

# Minor points
1. Fig 1 is misleading. (1) makes it sounds like all existing deep learning methods have this preprocessing approach, which is false. For example the Yu et al reference "Remote Photoplethysmograph Signal Measurement from Facial Videos Using Spatio-Temporal Networks" (which in the References section should be updated to BMVC 2019) only assumes preprocessing with a (very lightweight and commonly available) Viola-Jones face detector. It should be made clear that (1) refers to specific approaches, not *all* existing approaches.

2. The use of bolding in Table 1 is abused a bit, because EfficientPhys-C is bolded even when it is not the best result for the given metric. Stating that "we argue that the margin is relatively small as both Dual-GAN and EfficientPhys-C achieve a Pearson correlation of 0.99" does not justify bolding the values for EfficientPhys-C for MAE and RMSE, because this is a subjective assessment and the reader should be allowed to make their own subjective assessment.

3. Fig 2: N does not appear to be defined anywhere in the paper?

4. What are the limitations of this method? Based on Table 1, it appears that access to training data *might* be one which limits the transformer approach. Could another be that the difference layer only operates across contiguous frames? If the differences across contiguous frames were sufficiently small that they were drowned out by quantization noise, there might still be detectable differences across longer time periods. In such a case it might help to aggregate infomation across bigger frame gaps before throwing it away. It would be nice to see some discussion of the limitations of the approach here?

5. The paper mentions “second forward difference” but isn’t this a 1st order difference? i.e. torch.diff(n=1) rather than torch.diff(n=2)? second forward difference means x_{t+2} - 2x_{t+1} + x_t?

6. “the task of heart rate estimation... is considered as the gold-standard benchmark for the field of camera-based vital measurement” -- what is meant by gold-standard benchmark? It seems to imply that other vital sign measurements are not as important? The last few sentences of Visual Transformers part of Sec 2 have a few other typos/grammar errors.

7. It is also interesting to me what the performance gap is when inverting the train and test datasets -- what is the gap when testing on synthetic data? The nice thing there is that synthetic data ground truth should contain no errors, and it would also allow more careful analysis of any algorithmic bias.

8. The first paragraph on p4 “Normalization Module” describes related work, so probably belongs in a different section.

9. Table 1. Are the results here based on one training run per model? It is useful to understand the variation in performance per model when starting from multiple random seeds.


**Summary Of The Paper:**

This work proposes and contrasts two new approaches to estimate a photoplethysmography signal from facial videos using a supervised CNN- and transformer-based model.
Both models use a batch-normalized difference image input; the CNN model introduces the use of self-attention masks as opposed to more commonly used appearance-based attention masks, and the transformer model is based on the SwinTransformer model with tensor-shift modules to propagate information temporally.
The approaches are designed towards real-world deployment, by eliminating any pre-processing requirements and reducing on-device latency.
The models are evaluated in a cross-dataset setting, training on a combined real and synthetic dataset, and testing on three separate held-out real datasets. This evaluation shows that the proposed approaches, in particular the convolutional approach, perform well.
The accuracy vs latency trade-off is also assessed, showing that the approach can be made to run relatively efficiently on a Raspberry Pi Model 4B.


**Summary Of The Review:**

I thought this paper was generally well written and I appreciate the overall goal of it, which is to make camera-based measurement of certain vital signs more easily accessible and therefore useful. I also appreciate the effort to apply and adapt a new technique (vision transformers) to the problem, as well as the benefits of adjustments to TS-CAN to improve its performance. While the results on the cross-dataset experiment look strong, I find the technical contributions and analysis to be a little light for the level I would expect of ICLR and I would have preferred to have seen more careful study of a single method to understand its benefits. For example, if focused on EfficientPhys-C, what are the effects of the changes to the model vs TS-CAN? Or if focused on EfficientPhys-T, what other architectural alterations could be explored to improve latency? I think that the approach introduced in the paper is good and that with some additional care towards framing, explanation and analysis, this can be a nice paper and contribution to the physiological monitoring literature.

---

### Public Comment · ~Amogh_Gudi1 · 2021-11-10
**Missing: Evaluation on wilder/larger datasets + Comparison with prior work + Efficiency study on all datasets**

Great work and topic. It is nice to see a focus on efficiency in the field of rPPG.

This is a review of the **evaluation** done in the paper:

1. Experiments: The 3 evaluated datasets are relatively small and under lab recording/lighting condition and little movement. More robust evaluation can be performed by using larger-scale dataset with more realistic recording condition. E.g., VIPL-HR [A] with 2000+videos of 100+subjects with multiple movement, lighting and camera types; MoLi-PPG [B] with 30 subjects and systematic variations in lighting and movement; etc.
2. Experiments: It would be good to mention the window size used for the evaluation (for computing HR). This doesn't seem to be mentioned anywhere, but it can affect results strongly (as discussed in [C,E]). For a fair comparison, all methods should should use the same window size (if not, perhaps this should be mentioned).
3. Comparison with SOTA, Table 1: [D-K] also publishes results on some/all datasets used in the paper. It would be better if they are also included in the comparison. It looks like some perform better/equal.
4. Related work: Are there no other papers that focus on efficiency for rPPG (apart from MTTS-CAN)? Some papers that also focus on real-time/efficient rPPG implementation are [D,E]. The related work text could benefit from a fresh literature survey that lists all relevant work.
5. Computational cost, Table 2, Figure 4: [D/E] also mentions computational costs/latency. For context, it would make sense to also add their results to the efficiency study as their reported accuracy vs latency appears to be quite close to the proposed method.
6. Computational cost, Figure 4: It seems that efficiency is only studied on UBFC, while PURE and MMSE have been skipped. The results would appear more robust if efficiency results on all evaluated datasets are shown, or at least an average (weighted) over all datasets. This would also dispel doubts about cherry-picking of datasets.

References:
- [A] Niu, X.; Han, H.; Shan, S.; Chen, X. VIPL-HR: A Multi-modal Database for Pulse Estimation from Less-Constrained Face Video. In Lecture Notes in Computer Science; 2019; Vol. 11365 LNCS, pp. 562–576.
- [B] Artemyev, M.; Churikova, M.; Grinenko, M.; Perepelkina, O. Robust algorithm for remote photoplethysmography in realistic conditions. Digital Signal Processing 2020, p. 102737.
- [C] Mironenko, Y., Kalinin, K., Kopeliovich, M., & Petrushan, M. (2020). Remote photoplethysmography: Rarely considered factors. In Proceedings of the IEEE/CVF Conference on Computer Vision and Pattern Recognition Workshops (pp. 296-297).
- [D] Gudi, A.; Bittner, M.; Lochmans, R.; van Gemert, J. Efficient Real-Time Camera Based Estimation of Heart Rate and Its Variability. Proceedings of the IEEE International Conference on Computer Vision Workshops, 2019.
- [E] Gudi, A., Bittner, M., & van Gemert, J. (2020). Real-Time Webcam Heart-Rate and Variability Estimation with Clean Ground Truth for Evaluation. Applied Sciences, 10(23), 8630.
- [F] Macwan, R.; Benezeth, Y.; Mansouri, A. Heart rate estimation using remote photoplethysmography with multi-objective optimization. Biomedical Signal Processing and Control 2019, 49, 24–33.
- [G] Macwan, R.; Bobbia, S.; Benezeth, Y.; Dubois, J.; Mansouri, A. Periodic variance maximization using generalized eigenvalue decomposition applied to remote photoplethysmography estimation. Proceedings of the IEEE Conference on Computer Vision and Pattern Recognition Workshops, 2018, pp. 1332–1340.
- [H] Song, R.; Zhang, S.; Cheng, J.; Li, C.; Chen, X. New insights on super-high resolution for video-based heart rate estimation with a semi-blind source separation method. Computers in biology and medicine 2020, 116, 103535.
- [I] Yu, Z.; Li, X.; Niu, X.; Shi, J.; Zhao, G. AutoHR: A Strong End-to-End Baseline for Remote Heart Rate Measurement With Neural Searching. IEEE Signal Processing Letters 2020, 27, 1245–1249. doi:10.1109/LSP.2020.3007086.
- [J] Yu, Z.; Li, X.; Zhao, G. Remote Photoplethysmograph Signal Measurement from Facial Videos Using Spatio-Temporal Networks. 30th British Machine Vision Conference 2019, 2019, p. 277.
- [K] Artemyev, M.; Churikova, M.; Grinenko, M.; Perepelkina, O. Robust algorithm for remote photoplethysmography in realistic conditions. Digital Signal Processing 2020, p. 102737.

Hope you find these comments useful! :)

---

### Decision · Program_Chairs · 2022-01-20

**Decision:**

Reject

**Comment:**

The paper proposes a model for heart rate estimation from video. Unlike previous approaches, the method does not perform pre-processing of the video (face detection, cropping, etc). The empirical results are good - on par with state of the art or sometimes better.

Most reviewers are negative even after considering the authors' responses, but there is no full consensus. Some mentioned pros of the paper are that it's practically useful to have a model without pre-processing steps and that overall the technical side of the paper seems sound. Some cons are that the technical novelty is limited, the empirical evaluation is somewhat limited, and appropriate ablation studies are missing.

Overall, I recommend rejection at this point. While the paper goes in a promising direction, it is an application paper and as such it would be expected to have a more solid experimental evaluation - ideally on multiple tasks (as the paper title might suggest) and on more datasets as mentioned by the reviewers and the commenters. Moreover, an ablation study (and/or other analysis) of the proposed model would be crucial to let the readers know what components of the model actually contribute to its performance.